# Endometriosis-Associated Ovarian Cancer: The Origin and Targeted Therapy

**DOI:** 10.3390/cancers12061676

**Published:** 2020-06-24

**Authors:** Kosuke Murakami, Yasushi Kotani, Hidekatsu Nakai, Noriomi Matsumura

**Affiliations:** Department of Obstetrics and Gynecology, Kindai University Faculty of Medicine, Osakasayama, Osaka 589-8511, Japan; kmurakami@med.kindai.ac.jp (K.M.); y-kotani@med.kindai.ac.jp (Y.K.); nakai@med.kindai.ac.jp (H.N.)

**Keywords:** chocolate cyst, clear cell carcinoma, endometrial cyst, endometrioid carcinoma, endometriosis, endometriosis-associated ovarian cancer, malignant transformation

## Abstract

Endometrial cysts (ECs) are thought to be the origin of endometriosis-associated ovarian cancer (EAOC). A hypothesis that the oxidative stress of iron in cysts causes “malignant transformation of ECs” has been proposed, but this has not been verified. Several population-based studies showed that endometriosis was a risk factor but did not reflect the “malignant transformation of ECs”. A review showed that most patients were diagnosed with EAOC early in monitoring following detection of ECs, and that these cases might have been cancer from the start. Epidemiologically, EAOC was reduced by hysterectomy rather than by cystectomy of ECs. Gene mutation analyses identified oncogenic mutations in endometriosis and normal endometrium and revealed that the same mutations were present at different endometriotic lesions. It was also shown that most of the gene mutations found in endometriosis occurred in normal endometrium. Taking together, EAOC might be caused by eutopic endometrial glandular epithelial cells with oncogenic mutations that have undergone menstrual blood reflux and engrafted in the ovary, rather than by low-risk ECs acquiring oncogenic mutations and becoming malignant. This review discusses the mechanisms of EAOC development and targeted therapy based on genetic variation in EAOC with a focus on eutopic endometrium.

## 1. Introduction

Endometriosis is a disease in which the endometrium or similar tissue develops outside the original uterine cavity [1]. Endometriosis is a risk factor for ovarian cancer [2], and in a recent meta-analysis, the odds ratio for ovarian cancer in the presence of endometriosis was 1.42 [3]. Endometriosis-associated ovarian cancer (EAOC) is thought to develop from ovarian endometrial cysts (ECs) [4]. EAOC is mainly endometrioid or clear cell carcinoma. Ovarian endometrioid carcinoma (OEC) is the most common type of EAOC, occurring in approximately 75% of cases [5]. In Japan, however, ovarian clear cell carcinoma (OCCC), which is refractory to chemotherapy and has a poor prognosis, is more prevalent, accounting for approximately a quarter of all epithelial ovarian cancers [6].

Regarding the management of endometriosis, the European Society of Human Reproduction and Embryology (ESHRE) guidelines recommend that even if endometriosis increases the risk of developing ovarian cancer, there is no way to reduce it, and therefore its management should not be changed with concern for the ovarian cancer [7]. However, the Japanese guidelines still emphasize the risk of the malignant transformation of ECs, and the concept of “prevention of malignant transformation” is presented. Especially in patients over 40 years of age, oophorectomy for the purpose of histological retrieval is considered [8]. Therefore, the treatment of ECs is controversial when symptoms are negligible without treatment or when symptoms are under control with medication.

In this review, we discuss the mechanisms of EAOC development and targeted therapy according to genetic variations in EAOC based on the evidence to date, with a focus on eutopic endometrium.

## 2. Endometriosis and EAOC

Endometriosis is thought to involve the reflux of menstrual blood into the abdominal cavity [4]. Increased frequency of menstruation makes endometriosis more likely to arise [9]. As a result, it has been reported to occur in approximately 10% of women of reproductive age in modern society, where the prevalence of late births is increasing [10]. The main symptoms of endometriosis are dysmenorrhea and infertility [1]. Other symptoms in patients with endometriosis include lower back pain, lower abdominal pain, defecation pain, and sexual intercourse pain, even if they are not menstruating. These symptoms are more common in women in their 20s and 30s and abate with a decrease in estrogen as they age [10]. Treatment of endometriosis is mainly medication or surgery and is selected based on the type and severity of the symptoms, as well as the patient’s age and wishes for pregnancy. Medications are effective for pain, and when analgesics, such as non-steroidal anti-inflammatory drugs (NSAIDs), are ineffective, oral contraceptives (OCs) or progestin may be used to relieve symptoms by suppressing the secretion of estrogen or acting directly on the lesion [7]. Gonadotropin-releasing hormone agonists and antagonists may also be used when patients fail to respond to OCs or progestin [7,11]. Thus, with the recent developments in therapies, surgery can often be avoided in many cases of ECs [12]. However, a major problem is that ovarian cancer can develop during follow-up of tumors that are considered as ECs.

There have been many reports of oxidative stress in the microenvironment of endometriosis [13]. Oxidative stress in endometriosis is thought to cause infertility and carcinogenesis [4,14]. Yamaguchi et al. reported that fluid in ECs contains abundant iron, which causes oxidative stress; 5-fold diluted fluid from ECs caused DNA damage in Chinese hamster lung fibroblast V79 cells [15]. Excess iron generates reactive oxygen species (ROS) through the Fenton reaction (Fe^2+^ + H_2_O_2_ → Fe^3+^ + OH^−^ + •OH) [16]. ROS causes not only gene mutations but also epigenetic alterations, such as DNA methylation [17]. Kobayashi et al. proposed a hypothesis suggesting that at least two steps are necessary for ECs to develop into EAOC [18]. In the first step, DNA damage, mutations, and genomic instability might be caused via the Fenton reaction. High levels of ROS might be cytotoxic and cause apoptosis, and thus may not promote carcinogenesis. In the second step, when cells with high antioxidant capacity arise, selection of these cells might occur, resulting in their proliferation to cause carcinogenesis. However, there has been no data on whether EAOC has a strong ability to resist oxidative stress, and there is no evidence to support this hypothesis. In addition, it has recently been reported that the interaction between oxidative stress and non-coding microRNAs has an important role in the development of EAOC [19], but further study is required.

## 3. Clinical Epidemiology

### 3.1. How Long Does It Take for ECs to Become EAOC?

Women with endometriosis are more likely to develop EAOC in their late 40s, but after menopause, the incidence of ovarian cancer is not increased [20,21]. The risk ratio of ovarian cancer after menopause is 0.8 in women with previously diagnosed endometriosis [22]. There is no evidence that EAOC is more likely to develop in older women or that carcinogenesis is increased if ECs are followed up after menopause. These data contradict the hypothesis that prolonged exposure to the contents of ECs leads to DNA damage and increases carcinogenesis. Therefore, this hypothesis is not supported by the epidemiological data.

In Japan, a prospective cohort study of carcinogenesis during follow-up of ECs was performed [23]. This study enrolled 6398 cases of EC at an average age of 38.4 from 1985 to 1995 and followed up for an average of 12.8 years. As a result, ovarian cancer was diagnosed in 46 cases (0.7%). In this study, the risk ratio was adjusted for age at the time of diagnosis and compared with healthy women. Women with ECs were nine times more likely to have ovarian cancer than healthy women. The histological types were mostly clear cell carcinoma and endometrioid carcinoma. Although it was published 12 years ago, it is still the only report to examine the risk of carcinogenesis when following up cysts diagnosed as ECs. However, these results must be interpreted with care. Firstly, the small sample size requires caution in interpretation regarding the frequency of 0.7% and the figure of 9 times compared to the control. Secondly, the study was begun more than 35 years ago when transvaginal ultrasound and MRI were uncommon, and hence the diagnostic accuracy might not reflect the present diagnostic capabilities. Thirdly, this study did not distinguish between carcinogenesis from the observed ECs and that derived elsewhere. Hence, this study examined “the risk of women who had been diagnosed with ECs being subsequently diagnosed with ovarian cancer”, rather than “malignant transformation of ECs”.

We reviewed case reports of ovarian cancer originating from ECs that have been published since 2000 because these cases were likely to be well observed by transvaginal ultrasound or MRI [24]. In total, 79 cases in 32 articles were collected, and the histological types were mostly clear cell carcinoma and endometrioid carcinoma. The authors were all Japanese except for two papers (two cases). The cumulative carcinogenesis rate was 50% at 3 years and 75% at 5 years. Except for one case in the 13th year, all cases were diagnosed with cancer within 10 years. The number of cases of carcinogenesis was highest in the first year after the start of follow-up and tended to decrease over time. These results were very surprising, as it was expected that there would be a publication bias towards cases that developed cancer after long-term follow-up. Adenocarcinoma takes more than five years to reach a detectable size after cancer cells develop [25]. In other words, it was considered that almost all reported cases of “malignant transformation of ECs” might be “cancer from the beginning”.

### 3.2. Probability of Ovarian Cancer Being Diagnosed in Surgical Specimens of Endometrial Cysts

When surgery is performed on patients with a diagnosis of ECs, ovarian cancer may be identified pathologically. In Taiwan, a retrospective single-center study found that 7629 surgeries were performed following preoperative diagnosis of ECs and 0.14% (11 cases) were diagnosed with ovarian cancer postoperatively [26]. If such cases were followed without surgery, ovarian cancer might have developed within a few years.

In another study, among patients diagnosed with endometriosis in the Taiwan National Health Insurance Research Database who underwent surgery from 2000 to 2010, ovarian cancer occurred in 39/5945 cases in the endometriosis group and 36/23,780 cases in the control group [27]. The hazard ratio (HR) of ovarian cancer with endometriosis was 5.6 and that of developing OCCC was 7.4. However, the actual cumulative curve for carcinogenesis demonstrated that 0.4% of cases had been diagnosed with ovarian cancer within a month of being registered as endometriosis (Figure 1). That is, it was considered that the “endometriosis group” included 0.4% (24 cases) of “ovarian cancer”. If these cases had been followed up without surgery, many would have been diagnosed with cancer within a few years. Except for the first month after enrollment, 12 cases of ovarian cancer occurred after surgery for endometriosis, but the risk was approximately 2–3 times higher than that of the controls. This is greater than the data of the population-based study, which is going to be discussed below, but might be because the median age of the cohort was around 40 years and the analysis was limited to the age at which the risk ratio of ovarian cancer was greatest with endometriosis.

### 3.3. Large-Scale Population-Based Studies

Among 13 case-control studies, the relationship between endometriosis and the risk of developing ovarian cancer was investigated in 7911 ovarian cancers and 13,226 controls [28]. Twelve of the 13 studies were population based, with the earliest case collection beginning in 1992. When there had been a history of endometriosis, the risk of ovarian cancer was 1.49 times greater, with the risk of OCCC and OEC significantly higher at 3.73 times greater and 2.32 times greater, respectively. Low-grade serous carcinoma, unrelated to endometriosis, also increased by 2.02 times. Seromucinous borderline tumor, which is similar to serous borderline tumor and low-grade serous carcinoma at low magnification, is known to be strongly associated with endometriosis, but this pathological concept was clearly presented in 2002 [29]. Thus, a certain number of tumors that are now classified as seromucinous borderline tumor might have been classified as low-grade serous carcinoma in the above case-control studies. In a meta-analysis of 22 large population-based cohorts of 1.3 million women, 5584 cases of ovarian cancer were diagnosed and were examined for risk factors [30]. The risk of ovarian cancer was shown to increase by 1.35 times if previously diagnosed with endometriosis. The risk increased by 2.87 times in OCCC and by 2.32 times in OEC. The reason for the lower risk ratio in this study might be that 17 of 22 cohorts had a median age of 50 years or older. The method of the large-scale population-based study as described above showed that a history of endometriosis was a risk factor, rather than “malignant transformation of ECs”. In such population-based studies, “cancer from the beginning” is less likely to be included.

A meta-analysis was conducted on the risk of developing ovarian cancer in women with endometriosis [31]. However, this contained various methods of research and was difficult to interpret. 

### 3.4. Surgery and Medication for Endometriosis and Risk of Ovarian Cancer

To determine the origin of cancer, it is useful to identify precancerous lesions by histopathology. The histology of ECs often shows a simple monolayer of epithelium, with few abnormal mitotic cells or stratification. In many cases, this monolayer of epithelium and endometrial glandular epithelial cells are hardly found [32]. That is, most EC cases do not have precancerous lesions. It has been reported that “atypical endometriosis”, which is assumed to be a precancerous lesion, was found in 60% of cases of endometriosis with ovarian cancer but in only 2% of cases without ovarian cancer [33]. However, the interpretation of these data requires caution, as there are very few reports on atypical endometriosis and its definition has not been determined. Nuclear atypia with no proliferative features, which are often seen, is considered to be a reactive change. A proliferative lesion similar to a precancerous lesion of endometrioid carcinoma of the endometrium is considered to be a true precancerous lesion, but this is rarely observed [34].

The relationship between the type of surgery and the risk of developing ovarian cancer is also important. Fallopian tubal ligation reduces serous carcinoma by only about 20% but OCCC and OEC by about 50%. The effect is observed early (2–9 years) after tubal ligation and continues thereafter [35,36].

If ECs are the origin of ovarian cancer, cystectomy should be able to prevent ovarian cancer. However, it was reported that ovarian cancer sometimes occurs in the ipsilateral ovary following cystectomy [37,38]. The risk ratio of developing ovarian cancer after surgery (most likely cystectomy or oophorectomy) for endometriosis compared with controls was 1.69 in Denmark (3.37 for OEC and 3.03 for OCCC) [39], 1.77 in Scotland [40], and 1.60 in Canada [21]. That is, in such surgery, the risk of developing ovarian cancer did not decrease as compared with the population-based data described above. However, according to a report from Sweden, in women hospitalized for a diagnosis of endometriosis, the risk ratio of ovarian cancer was 1.54 without hysterectomy and 1.05 with hysterectomy, with hysterectomy eliminating the risk of ovarian cancer [20]. In addition, in a study reported in 2019 that followed 830,000 women for 27 years, hysterectomy without oophorectomy reduced the risk of ovarian cancer to HR 0.17 in women with endometriosis [41]. Thus, the risk of developing EAOC was reduced with hysterectomy rather than cystectomy of ECs. Furthermore, as described above, there was no increase in the incidence of ovarian cancer after menopause related to a history of endometriosis. These results suggest that the reflux of eutopic endometrial glandular epithelial cells through the fallopian tube might be involved in the development of EAOC, and not through the endometriotic lesions gradually turning into cancer. When using OCs for more than 10 years, the risk ratio for ovarian cancer was 0.21 with endometriosis and 0.47 without endometriosis [42]. Therefore, a reduction in menstrual blood and suppression of ovulation by OCs might also contribute to the prevention of EAOC.

## 4. Genetic Analysis of Endometrium, Endometriosis, and EAOC

It has been suggested that EAOC arises from endometriosis because EAOC exhibits mutations in *ARID1A*, *PIK3CA*, and *PTEN* as well as loss of heterozygosity at a high frequency, which is also common in coexisting endometriosis [43,44,45,46]. Recently, however, more detailed genetic mutation analyses of endometriosis using next-generation sequencing (NGS) are changing the conventional concept. In reports examining deep endometriosis, which is not usually associated with carcinogenesis, 19 of 24 patients had gene mutations and 5 had oncogenic mutations. Endometriotic lesions in these cases, such as in the Douglas pouch, rectal surface, and peritoneum of the vesicouterine pouch, carried the same gene mutation [47]. In other words, it was presumed that the origin of these gene mutations was eutopic endometrium. In a study of 107 cases of EC and 82 cases of normal endometrium, Suda et al. reported numerous *KRAS* and *PIK3CA* mutations in both endometriosis and normal endometrial glandular epithelium, and the same mutations were observed even at different sites of endometriosis [48]. Similarly, it has been reported that many of the genetic mutations found in endometriosis can also be found in eutopic normal endometrium by exome sequencing [49]. In mouse models of *ARID1A* and *PIK3CA* mutations in the endometrium, it was reported that endometriosis occurred when the endometrial glands were easily regurgitated into the peritoneal cavity by salpingectomy [50]. It has also been reported that eutopic endometrium has numerous genetic mutations, the number of which is correlated with age [51,52].

Focusing on the analysis of EAOC, it was reported that EAOC and nearby endometriosis were the same clone [53]. Notably, the endometriosis of the Douglas pouch, probably caused by reflux from the uterus, also had the same clone as the EAOC. This result suggested a relationship between EAOC and eutopic endometrium.

The relationship between endometrial cancer, especially endometrioid carcinoma, and endometriosis has also been reported. Epidemiologically, the risk of developing endometrioid carcinoma of the endometrium is as high as 2.1 [54] or 2.8 times [55] in endometriosis cases. Furthermore, in cases of double cancer of the endometrium and ovary, endometriosis significantly coexisted, with 13 out of 13 cases [56] and 21 out of 24 cases [57]. Genomic sequencing of double cancer of the endometrium and ovary revealed they had the same clone in 22 of 23 cases except Lynch syndrome [58]. In another report, 13 out of 14 cases had the same clone [59]. Thus, endometrial cancer, endometriosis, and ovarian cancer may be strongly related to each other in double cancer.

A summary of the genetic alterations commonly found in endometrium, endometriosis, and EAOC is shown in Table 1. All of them are driver genes whose mutations were confirmed by sequencing. Many driver gene mutations are common to endometrium, endometriosis, and EAOC. In addition, when comparing these with endometrial cancer gene mutations from The Cancer Genome Atlas [60], a significant number of them are consistent with endometrium, endometriosis, and EAOC gene mutations. This suggests that EAOC is similar in nature to endometrial cancer arising from the endometrium.

## 5. Reconsideration of the Developmental Mechanism of EAOC Based on Molecular and Epidemiological Data

From the epidemiological and molecular data described above, there are several pieces of evidence that could not be well explained if ECs themselves had become EAOC. Therefore, we now consider the mechanism of EAOC carcinogenesis as follows (Figure 2). Carcinogenesis requires an initiator and a promoter. First, gene mutations occur frequently in endometrial glandular epithelial cells, which enables endometrial cells to survive when they flow back into the abdominal cavity. However, endometriosis is much more common than EAOC because its mutations are often insufficient as initiators for ovarian cancer. Nonetheless, eutopic endometrial glandular epithelial cells with sufficient gene mutations, when engrafted in the ovaries by menstrual blood reflux, cause carcinogenesis through the effect of the cancer-promoting ovarian microenvironment. However, cells with an initiator mutation may not always cause carcinogenesis, which is similar to the fact that inoculating immunodeficient mice with cancer cells does not result in 100% tumor formation. Therefore, even if the cells are from the same clone as the cancer, if they survive in the contralateral ovary or outside the ovary, they may not cause carcinogenesis, and such cells become endometriosis. Thus, EAOC with endometriosis occurs. In addition, the defect in the ovarian surface caused by ovulation may be particularly rich in growth factors as promoters, and OC prevents EAOC through the suppression of ovulation. The wound after cystectomy of ECs also causes the microenvironment to become rich in growth factors, and EAOC is likely to develop in the ipsilateral ovary after cystectomy.

In ovarian cancer, the prognosis for recurrence in the preserved ovaries after fertility-conserving treatment is good if radical surgery is performed again [61], and all six cases of EAOC (OEC and OCCC) have been reported to be cured [62]. That is, these recurrences may not be the result of hematogenous metastasis or intra-abdominal dissemination, but rather, the tumor cells may remain in the genital tract and be transported from the endometrium to the ovaries.

Our hypothesis can be naturally inferred from epidemiological and underlying data. However, no data to directly prove this has been reported. Although many of the gene mutations in the glands of endometriosis tissue were also found in normal eutopic endometrium by exome sequencing [49], the study did not look for mutations at each gland. Endometrial glands are known to be monoclonal [53,63]; therefore, to prove that endometriosis or EAOC is caused by the regurgitation of endometrial gland ducts with genetic mutations, it is necessary to analyze each gene mutation in the glands and to identify the glands that are the same clones as endometriosis and EAOC. As demonstrated by gene mutation analysis that high-grade serous carcinoma is derived from serous tubal intraepithelial carcinoma (STIC) [64], if precancerous lesions are detected by meticulous sectioning of the endometrium in EAOC, it is considered that the sequencing analysis can prove that EAOC is derived from the endometrium. However, it is considered difficult to select endometrial glands to be sequenced based on histopathology. To prove that eutopic endometrium with genetic mutations causes endometriosis and EAOC, it is necessary to further advance the sequencing technology in addition to meticulous sectioning of the endometrium.

In addition, the question remains as to why clear cell carcinoma rarely occurs in the endometrium. Normal endometrial glandular epithelium includes estrogen receptor (ER)-positive secretory cells and ER-negative ciliated cells. Recently, it was proposed that each of them is highly likely to be the origin of OEC and OCCC, respectively [65]. Since there are few ER-negative ciliated cells in the eutopic endometrium, the proliferation of ciliated cells might be suppressed in the endometrial environment through interactions with stromal cells. The reason that clear cell carcinoma is more common in the ovary but less so in the endometrium might be because the mechanism that suppresses ciliated cell proliferation does not work in the ovarian microenvironment but rather is advantageous for ciliated cell proliferation. This hypothesis needs to be verified by basic research in the future.

Regarding endometriosis and ovarian mesenchymal tumors, it has been reported that endometrioid stromal sarcoma and adenosarcoma are associated with endometriosis. They are both extremely rare tumors. In a case series of 63 cases of endometrioid stromal sarcoma, 27 cases had ovarian involvement, and 16 of the 27 cases also had endometriosis [66]. However, detailed histopathology of the endometriosis and tumors in each case was not presented and a genomic analysis was not performed. As for adenosarcoma, it often results from extraovarian endometriosis [67], but only a few cases have been reported that suggested an association with ECs [68]. Thus, the pathogenetic mechanism of ovarian mesenchymal tumor development is unclear, and further study is required.

## 6. The Pros and Cons of Managing Endometriosis with Ovarian Cancer Development in Mind

The World Endometriosis Society stated that the relative and absolute risk of ovarian cancer in women with endometriosis is very low and routine screening for ovarian cancer was not recommended [70]. The ESHRE guidelines also recommended not to manage endometriosis for carcinogenesis [7]. ECs cause infertility, dysmenorrhea, and abdominal pain due to infection or rupture. Furthermore, differentiation from ovarian cancer is difficult in some cases. Therefore, surgery is required after all in many individuals. However, in ECs, there is no evidence for screening to detect cancer at early stage, or for surgery aimed solely at preventing carcinogenesis.

If it is not necessary to consider fertility preservation when performing surgery for symptoms or other gynecological disorders, the risk of ovarian cancer associated with endometriosis may be reduced by tubal ligation, salpingectomy, and hysterectomy. Considering the prevention of high-grade serous ovarian carcinoma, which is known to originate from the fallopian tubes, salpingectomy might be better than tubal ligation [71]. OCs reduce the risk of ovarian cancer. However, it does not mean that OCs prevent the “malignant transformation of ECs”. If the ECs were actually “cancer from the beginning”, it is unlikely that OCs would prevent ovarian cancer. There is no evidence that asymptomatic ECs should be followed up. It might be a good idea to follow up several months or one year after diagnosing “ECs” to identify “cancer from the beginning” by monitoring the size or solid region. However, if the size decreases after menopause, it is unlikely that the cyst is “cancer from the beginning”, and the risk of developing ovarian cancer may not be higher, so it is not necessary to monitor. Screening of ovarian cancer has no effect on slow-growing type I ovarian cancer, including EAOC [72]. Ovarian cancer occurs in around 1–2% of women as long as the ovaries are present, but screening for ovarian cancer is more likely to be harmful to the patient as a result of the unnecessary medical care and thus should be avoided [73].

If we can accurately and noninvasively diagnose whether a tumor is EC or EAOC, we can avoid overdiagnosis and overtreatment resulting from concerns regarding the malignant transformation of ECs. It has been reported that many ovarian cancer cases have gene mutations and chromosome number abnormalities (copy number increases of chromosome 7q and 8q) in cervical Pap smear samples or endometrial Tao brush samples [74]. Even when limited to stage I OCCC or OEC, findings were positive in 10 of 23 cervical Pap smear samples and in two of four endometrial Tao brush samples. This may be because of the detection of genetic mutations in endometrial glandular epithelial cells or cell-free DNA derived from these cells. Therefore, if an ovarian cyst is identified that appears to be an EC, it may be accurately diagnosed as ovarian cancer by examining a sample from the cervix or endometrium for genetic abnormalities. Furthermore, proteomic analysis of endometrial fluid and circulating tumor DNA may be used to detect precursor lesions in EAOCs and to investigate the risk of developing EAOCs [75].

## 7. Landscape of Targeted Therapy and Immunotherapy for EAOC

There is still no standard use of therapies targeting EAOC genetic mutations. However, the development of targeted therapy will play an important role in improving the prognosis of OCCC, especially in chemotherapy-resistant OCCC. In addition, immunotherapy has been shown to be effective as a treatment for EAOC.

In OCCC, *PIK3CA* mutations are found in 33–40% of cases [76], activating the PI3K/AKT/mTOR pathway. Many clinical trials have been conducted on inhibitors of this pathway, including PI3K, AKT, and mTORC1 inhibitors [77]. None have yet been used as a standard treatment, but they could be a promising targeted therapy. A study that performed whole exome sequencing in 48 OCCC cases found several patients that may have responded to existing molecular targeted drugs; furthermore, this study demonstrated the utility of OCCC in exome sequencing [78]. Poly (ADP-ribose) polymerase (PARP) inhibitors have been shown to be effective against ovarian cancer, with *BRCA1/BRCA2* gene mutations resulting in homologous recombination deficiency [79,80,81,82]. Although *BRCA1/2* gene mutations resulting in homologous recombination deficiency are found in only 6–8% of EAOCs [83], PARP inhibitors may be effective in cases with mutations.

The expression of VEGF, an angiogenic factor, is significantly elevated in EAOC compared with endometriosis [84] and is reported to be expressed in more than 90% of OCCC cases [85]. Anti-VEGF antibodies, which are angiogenesis inhibitors, have already been used for ovarian cancer, including EAOC [86,87,88,89,90]. Although there are no data showing that anti-VEGF antibodies are more likely to be effective in EAOC than other histology types, it is possible that they may be shown to be effective for EAOC in combination with other drugs in the future.

Several clinical trials of immunotherapy for ovarian cancer have also been conducted, mainly with immune checkpoint inhibitors. In our phase 2 clinical trial of the anti-PD-1 antibody, nivolumab, for platinum-resistant ovarian cancer, the overall response was 15%; 2 of 20 patients had a complete response, one of which was pathologically OCCC [91]. In addition, a subgroup analysis of a phase 2 clinical trial of the anti-PD-1 antibody, pembrolizumab, for advanced and recurrent ovarian cancer (Keynote 100) showed an objective response rate to OCCC of 16%, which was higher than other histopathological types [92]. The results of these trials suggested that OCCC, among other EAOCs, is more likely to respond to immunotherapy. A phase 2 study of nivolumab ± ipilimumab for OCCC (NCT03355976) is currently in progress [93]. In addition, in Lynch syndrome, germline mutations are found in *MLH1*, *MSH2*, *MSH6*, and *PMS2* and their associated parts among mismatch repair genes, and the cumulative morbidity rate of ovarian cancer is higher (10–12%) compared with the general population [94]. In a study that performed whole exome sequencing of OCCC, there were 3 (6%) of 48 suspected cases of Lynch syndrome, suggesting that Lynch syndrome may be associated with a certain frequency in OCCC [76], in which mismatch repair gene mutations lead to high frequency microsatellite instability, a susceptibility biomarker for immune checkpoint inhibitors [95]. Therefore, it is more likely that immune checkpoint inhibitors will be effective in OCCC associated with Lynch syndrome.

The IL-6/JAK/STAT pathway is activated in OCCC, and it was reported that IL-6 expression was an independent poor prognostic factor in OCCC [96]. Studies using cell lines have suggested that inhibitors of this pathway are promising for treatment [97]. It was also reported that administration of an anti-IL-6 antibody to an OCCC mouse model improved prognosis [98], indicating the potential for future clinical applications.

*ARID1A* gene mutations are found in as many as 50% of OCCC cases [99,100]. The activity of histone deacetylase 6 (HDAC6) has been shown to be closely associated with ovarian cancer with *ARID1A* mutations [101], and the efficacy of HDAC6 inhibitors and combination therapy with anti-PD-L1 antibodies has been demonstrated in an OCCC mouse model lacking the *ARID1A* gene [102], which may also have future clinical applications.

## 8. Conclusions

In recent years, basic and epidemiological data suggesting that genetic mutations in the eutopic endometrium may be responsible for EAOC have been accumulating. EAOC might not occur as a result of malignant transformation of glandular epithelial cells of low-risk ECs that will gradually accrue gene mutations over a long period but might be caused by eutopic endometrial glandular epithelial cells with sufficient oncogenic mutations that are refluxed to engraft in the ovary. That is, “ECs” in which EAOC occurs during follow-up is considered to be “cancer from the beginning”. There is no evidence that ovarian cancer arises from ECs, and management with “malignant transformation of ECs” in mind might not be recommended. Within EAOC, especially for chemotherapy-resistant OCCC, it is necessary to conduct research and develop targeted therapy, keeping in mind that endometrial glandular epithelial cells serve as a carcinogenic site.

## Figures and Tables

**Figure 1 cancers-12-01676-f001:**
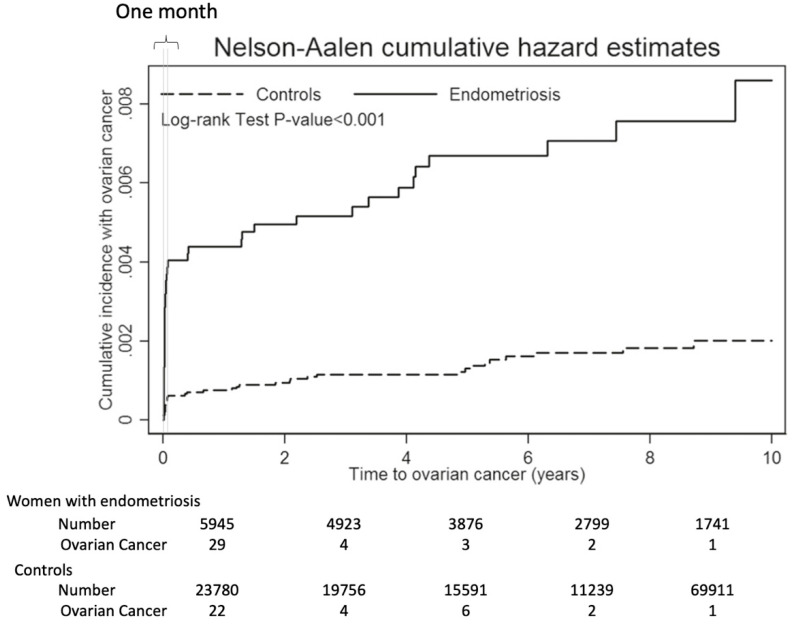
The actual cumulative curve for carcinogenesis from endometrial cysts. Graph showing that in 5945 women enrolled as endometriosis, 24 (0.4%) ovarian cancers developed within 30 days of enrollment. This figure was made by modifying a figure from BMC Cancer 2014 [27].

**Figure 2 cancers-12-01676-f002:**
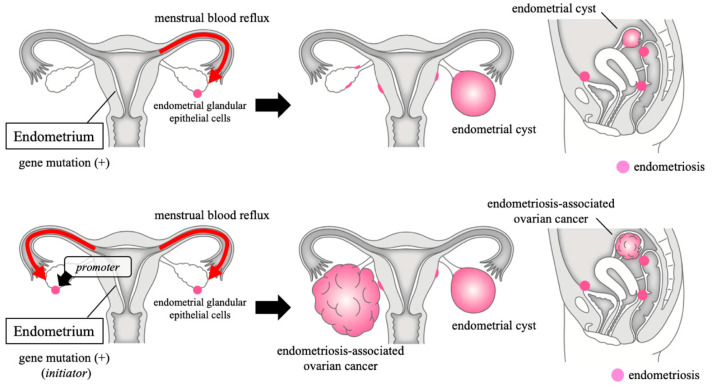
Mechanism of endometriosis-associated ovarian cancer carcinogenesis. Carcinogenesis requires an initiator and a promoter. Gene mutations occur frequently in endometrial glandular epithelial cells, but its mutations are often insufficient as initiators. Nonetheless, endometrial glandular epithelial cells with sufficient gene mutations cause carcinogenesis through the effect of the promoter. If they survive in the contralateral ovary or outside the ovary, they may not cause carcinogenesis, and such cells become endometriosis. This figure was made by modifying a figure from Clinical Gynecology and Obstetrics 2020 [69].

**Table 1 cancers-12-01676-t001:** Previous report of gene alteration in endometrium, endometriosis, endometriosis-associated ovarian cancer, and endometrial carcinoma.

Gene	Function	Specimen				TCGA
		Endometrium [48,51,52,103]	Endometriosis [47,48,103,104,105]	OCCC [76,78,99,100,103,106,107,108,109,110,111,112,113,114]	OEC [100,103,110]	Endometrial Carcinoma [60]
*PIK3CA*	PI3K/Akt/mTOR pathway	●	●	●	●	●
*PTEN*	PI3K/Akt/mTOR pathway	●	●	●	●	●
*ERBB2*	PI3K/Akt/mTOR pathway	●	●	●		●
*PIK3R1*	PI3K/Akt/mTOR pathway	●		●	●	●
*KRAS*	Ras/MAPK pathway	●	●	●	●	●
*NF1*	Ras/MAPK pathway	●		●	●	
*ARID1A*	SWI/SNF complex		●	●	●	●
*PPP2R1A*	Serine/threonine-protein phosphatase	●	●	●		●
*MLH1*	Mismatch repair protein	●	●	●		●
*CTNNB1*	Wnt/β-catenin signaling pathway		●	●		●
*KMT2D*	Histone methyltransferase	●		●		
*FBXW7*	Ubiquitin ligase	●			●	●

OCCC: Ovarian clear cell carcinoma, OEC: Ovarian endometrioid carcinoma, TCGA: The Cancer Genome Atlas.

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
