# Peer review of "Endometriosis-Associated Ovarian Cancer: The Origin and Targeted Therapy"

_cancers, 2020, doi:10.3390/cancers12061676_

Round 1

Reviewer 1 Report

Dear editor,

The manuscript “Endometriosis-associated ovarian cancer: The origin and targeted therapy (cancers-827121)” provides a comprehensive review of the current literature regarding the etiopathogenesis of endometriosis-associated ovarian cancer. Authors not only summarize the current body of knowledge on this issue, but draw hypothesis and conclusions from them. Mainly, authors propose that endometrium and not endometriotic cysts in the ovary are in the origin of endometriosis-associated ovarian cancer.

The points covered in the manuscript are of interest in current research in endometriosis-associated ovarian cancer. The paper is well written, English is appropriate and the questions addressed in the paper can help the endometriosis and endometriosis-associated ovarian cancer research community improve research and potentially guide medical decisions.

Nevertheless, I would recommend the paper for publication in “cancers” only after some points are taken into consideration and solved, what would be a minor revision:

Regarding style, Fenton reaction (Line 73) needs to be appropriately described, using superscripts for valences.

Regarding references, I would suggest including more recent published reviews on the role of oxidative stress, endometriosis and endometriosis-associated ovarian cancer.

In lines 86 to 90, authors describe that risk of ovarian cancer is not increased after menopause, and this contradicts “the hypothesis that prolonged exposure to the contents of ECs leads to DNA damage and increases carcinogenesis”. When women reach menopause, the cyclic intracystic bleeding is interrupted, as it is the oxidation of haemoglobin and the oxidative stress mediated by the Fenton reaction. Therefore, I am not sure if I can agree/completely understand the sentence stated by the authors in its current form.

Considering Figure 1, which has been adapted from reference 25, authors describe a cohort of 5,945 patients, of whom 39 developed ovarian cancer. This represents a 0.65%, albeit the plotted line for the endometriosis group reaches a 0.8% at 10 years of follow-up. Please, reconsider the values of the y-axis.

The authors propose a new mechanism for endometriosis-associated ovarian cancer carcinogenesis, by which gene mutations in the endometrium act as initiating factor and defects in ovarian surface and associated growth factors act as promoters. Additionally, the authors state that several of the considered “malignant transformations of ovarian endometriomas” were actually cancer for the beginning. At this respect, do the authors consider that a minimally-invasive screening could be established in selected women by combining size and evolution of ovarian cysts and assessment of mutations present in endometrium and ovarian cancer but not in endometriosis (for instance, PIK3R1 or NF1) in endometrial tissue? If this is the case, please mention in the paper.

Considering both the hypothesis proposed by the authors and the alternation in ovulation of each ovary, might it be expected that over the time, women should develop bilateral asynchronic endometriosis-associated ovarian cancer?

In section 7 (“landscape of targeted therapy…”) the repertoire of clinical trials for targeted therapy and immunotherapy needs to be described in more detail. For instance, I would appreciate if authors could include a mention to the current clinical trials with immune-checkpoints inhibitors in endometriosis-associated ovarian cancer. In line 335-336, authors describe that “Anti-VEGF antibodies, which are angiogenesis inhibitors, have already been used for ovarian cancer, including 336 EAOC”, although they do not provide an appropriated reference. The same could be applied in line 333, “PARP inhibitors may be effective in cases with mutations” is this an affirmation made from the literature or is it a hypothesis from the authors? This needs to be clarified in the paper.

Author Response

Dear Editors and Reviewer 1,

Thank you very much for reviewing our manuscript and offering your valuable advice. We have addressed your comments with point-by-point responses and have revised the manuscript accordingly.

We will be grateful if you find our article suitable for publication. We look forward to hearing from you at your earliest convenience.

Yours sincerely,

Noriomi Matsumura, M.D., Ph.D.

Reviewer 2 Report

Journal Cancers (ISSN 2072-6694) Manuscript ID cancers-827121 Type Review Title Endometriosis-associated ovarian cancer: the origin and targeted therapy Authors Kosuke Murakami , Yasushi Kotani , Hidekatsu Nakai , Noriomi Matsumura  

REVIEWER REPORT

The study seems original, really.

Authors reviewed case reports of ovarian cancer originating from ECs that have been published since 2000: seventy-nine cases in 32 articles were collected, and the histological types were mostly clear cell carcinoma and endometrioid carcinoma. Moreover, in this review, Authors discuss the mechanisms of EAOC development and targeted therapy according to genetic variations in EAOC based on the evidence to date, with a focus on eutopic endometrium. Gene mutation analyses identified oncogenic mutations in endometriosis and normal endometrium and revealed that the same mutations were present at different endometriotic lesions and that most of the gene mutations found in endometriosis occurred also in normal endometrium. Taking together, Authors proposed the concept that EAOC might be caused by eutopic endometrial glandular epithelial cells with oncogenic mutations that have undergone menstrual blood reflux and engrafted in the ovary, rather than by low-risk ECs acquiring oncogenic mutations.

I advice the following major revision:

- to add this recent reference, commenting about the importance of the

application of highly sensitive and specific novel molecular biomarkers, that could identify cases of endometriosis with oncogenic potential : Endometriosis and endometriosis-associated cancers: new insights into the molecular mechanisms of ovarian cancer development. Ecancermedicalscience. 2018; 12: 803. Published online 2018 Jan 25. doi: 10.3332/ecancer.2018.803. PMCID: PMC5813919. PMID: 29456620

- I would suggest updating the references and reporting the actual absolute risk difference as compared to those estimated for the general population.

- Please enrich your review, also consider the association between endometriosis and mesenchimal tumor and discuss about possible pathogenetic mechanisms

-  to check text and format

Author Response

Dear Editors and Reviewer 2,

Thank you very much for reviewing our manuscript and offering your valuable advice. We have addressed your comments with point-by-point responses and have revised the manuscript accordingly.

We will be grateful if you find our article suitable for publication. We look forward to hearing from you at your earliest convenience.

Yours sincerely,

Noriomi Matsumura, M.D., Ph.D.

Round 2

Reviewer 2 Report

Authors have adequately revised the paper

I retain it suitable for publication in Cancers journal